# Vancomycin for Dialytic Therapy in Critically Ill Patients: Analysis of Its Reduction and the Factors Associated with Subtherapeutic Concentrations

**DOI:** 10.3390/ijerph17186861

**Published:** 2020-09-19

**Authors:** Fernanda Moreira de Freitas, Welder Zamoner, Pamela Falbo dos Reis, André Luís Balbi, Daniela Ponce

**Affiliations:** Internal Medicine, Botucatu School of Medicine—UNESP, São Paulo State University Julio de Mesquita Filho, Distrito de Rubiao Junior, Botucatu P.O. 18618687, Sao Paulo, Brazil; ferdsmoreira@msn.com (F.M.d.F.); welder.zamoner@unesp.br (W.Z.); pamela_falbo@hotmail.com (P.F.d.R.); andre.balbi@unesp.br (A.L.B.)

**Keywords:** sepsis, acute kidney injury, dialysis, vancomycin, pharmacokinetic-pharmacodynamic

## Abstract

This study aimed to evaluate the reduction in vancomycin through intermittent haemodialysis (IHD) and prolonged haemodialysis (PHD) in acute kidney injury (AKI) patients with sepsis and to identify the variables associated with subtherapeutic concentrations. A prospective study was performed in patients admitted at an intensive care unit (ICU) of a Brazilian hospital. Blood samples were collected at the start of dialytic therapy, after 2 and 4 h of treatment and at the end of therapy to determine the serum concentration of vancomycin and thus perform pharmacokinetic evaluation and PK/PD modelling. Twenty-seven patients treated with IHD, 17 treated with PHD for 6 h and 11 treated with PHD for 10 h were included. The reduction in serum concentrations of vancomycin after 2 h of therapy was 26.65 ± 12.64% and at the end of dialysis was 45.78 ± 12.79%, higher in the 10-h PHD group, 57.70% (40, 48–64, 30%) (*p* = 0.037). The ratio of the area under the curve to minimal inhibitory concentration (AUC/MIC) at 24 h in the PHD group was significantly smaller than at 10 h (*p* = 0.047). In the logistic regression, PHD was a risk factor for an AUC/MIC ratio less than 400 (OR = 11.59, *p* = 0.033), while a higher serum concentration of vancomycin at T0 was a protective factor (OR = 0.791, *p* = 0.009). In conclusion, subtherapeutic concentrations of vancomycin in acute kidney injury (AKI) patients in dialysis were elevated and may be related to a higher risk of bacterial resistance and mortality, besides pointing out the necessity of additional doses of vancomycin during dialytic therapy, mainly in PHD.

## 1. Introduction

Sepsis is an important cause of acute kidney injury (AKI) in critically ill patients (over 50% of cases), and half of these patients require acute renal support (ARS) [1,2,3,4]; therefore, the adoption of measures that aim to reduce mortality, as well as the costs associated with treatment and hospitalization, remain a huge challenge.

Among these measures, early administration of antimicrobial in the appropriate dose is highly impactful [5,6,7]. However, the pharmacodynamics of antibiotic drugs in critically ill patients are greatly modified and poorly known due to alterations in their absorption, distribution, metabolism and elimination [8,9,10]. Another as yet unclarified question is the correct adjustment of the antimicrobial dosage in AKI patients and in dialytic therapy [4,11], aiming to avoid toxicity and subtherapeutic concentrations, which may contribute to higher mortality [12].

Vancomycin is a widely used drug in intensive care units (ICUs), as oxacillin-resistant Gram-positive cocci are of the main infectious etiological agents. Vancomycin is a 10–50% protein-bound glycopeptide with a molecular weight of 1485 Da and a distribution volume of 0.4–1 L/kg, which acts to inhibit cell wall synthesis in Gram-positive bacteria. Regarding its pharmacodynamics, vancomycin is an antimicrobial whose action is both time- and concentration dependent, and the best predictor of its activity is the ratio of the area under the concentration curve (AUC) to minimal inhibitory concentration (MIC) [13].

Due to its molecular weight and partial linkage to proteins, vancomycin is poorly removed by intermittent haemodialysis (IHD) utilizing low-flux dialyzers (kuf < 20 mmHg), justifying, in the past, its administration every 7–10 days [14]. However, with the frequent use of high-flux dialyzers, this dosing frequency of vancomycin is being re-evaluated, since 30–40% of the dose might be removed per session in patients in a scheme of chronic intermittent HD, with a likely increase in the amount removed by therapies of longer duration [15,16,17], such as prolonged HD (PHD) and continuous HD (CHD).

Thus, the objective of this study was to evaluate the reduction in vancomycin by different methods of dialysis (IHD and PHD) in critical patients with AKI associated with sepsis, through a pharmacokinetic approach and pharmacokinetic-pharmacodynamic modelling, in order to identify the factors related to subtherapeutic concentrations.

## 2. Methods

This was a prospective cross-sectional clinical study that included patients hospitalized in an ICU of the Brazilian University Hospital, over 18 years of age, with AKI and a clinical presentation suggestive of acute tubular necrosis associated with sepsis, in acute renal support (IHD or PHD) and on administration of vancomycin. The following groups were excluded from the study: pregnant women, patients who were not using vancomycin, those that presented AKI of other aetiologies and those in chronic renal replacement therapy (dialysis or renal transplant). Patients who had their dialytic session interrupted for clinical or technical reasons were also excluded. The research project was approved by the local Commission of Ethics in Research in March of 2015 (protocol 1,477,511). Free and informed consent was given in writing by each patient before the start of the study procedures.

Patients were allocated into three groups according to the type of dialytic treatment prescribed by the nephrology assistant team, based on the patient’s hemodynamic conditions and on the indications and contraindications of each dialysis method: IHD, 6-h PHD and 10-h PHD.

The prescription of vancomycin followed a protocol instituted by the institution’s Commission of Control of Infections Related to Health Assistance, being administered in an initial dose of 20 mg/kg and 15 mg/kg/day after haemodialysis session, since all patients utilized high-flux hemodialysis membranes [15]. The dosage was adjusted according to the serum levels of vancomycin, which were assayed on alternate days. 

Proportion machines (*Fresenius 4008*) and high-flux hemodialysis membranes (FX 60, FX 80 or FX 100), with surface areas of 1.8, 2.0 and 2.2 m^2^, respectively, were used in IHD and PHD sessions. The IHD sessions were carried out with blood flow of 250 to 300 mL/min; a dialysate flow of 500 mL/min and FX 60, FX 80 or FX 100 hemodialysis membranes for 4 h. The PHD session was carried out with a blood flow of 200 mL/min, a dialysate flow of 300 mL/min and an FX 80 hemodialysis membrane when the prescribed duration was 6 h and an FX 60 hemodialysis membrane when the prescribed time was 10 h.

During the sessions, patients were anticoagulated with 50–100 UI/kg of heparin in a bolus, and 500–1000 UI/hour in the remaining hours. When anticoagulation was contraindicated, the system was cleaned with 50–100 mL of 0.9% sodium chloride every 30 min during the procedure. Concentrations of bicarbonate (28–36 mEq/L), potassium 1–4 mEq/L, sodium (135–142 mEq/L) and calcium (2.5 or 3.5 mEq/L) in the dialysis bath were adjusted according to the patients’ individual requirements and examinations.

A serialized collection of blood samples (2 mL each) was carried out through a central venous catheter for the determination of serum concentrations of vancomycin before starting dialysis, 2 and 4 h after the beginning of therapy and at the conclusion of therapy. The previous vancomycin dose was received 24 h before the dialysis session. Figure 1 depicts the dialysis procedure, administration of vancomycin and sampling moments.

The collected blood was transferred to vacuum collection tubes (10 mL) containing Benton Dickinson (BD) gel and kept in styrofoam or in a refrigerator until the last collection of the day, when they were forwarded to the laboratory. There, serum was separated by centrifugation (3000 *g*, 20 min) and transferred to properly labelled 2-mL microcentrifuge tubes (Eppendorf). Serum samples were stored in a temperature-controlled freezer at −20 °C.

Vancomycin concentrations were determined using a liquid chromatographic (HPLC)-luorescence method as reported previously by Lopez et al. [18]. Blood samples were centrifuged within 4 h of collection and serum was stored frozen at −80 °C until assayed. The pharmacokinetic (PK) parameters volume of distribution (Vd) and elimination half-life (T1/2) were calculated for each patient assuming a 1-compartment model. The area under the curve (AUC) was determined using the trapezoidal rule to the last non-zero time point. The AUC from the last non-zero time point was determined by the pharmacokinetic method, concentration/k, where k is the elimination rate constant determined from log-linear regression of the terminal phase of the concentration-time profile. Table 1 shows the formulas utilized for the calculation of vancomycin pharmacokinetic (half-life time, volume of distribution (L), dialytic clearance of vancomycin (L/h) and % reduction in vancomycin).

Pharmacodynamic (PD) data are related to the MIC obtained by the antimicrobial susceptibility testing done for each pathogen in the Microbiology Laboratory after the strain documentation. The predictive index of vancomycin effectiveness is expressed by AUC 0-24/MIC ratio >400 [21,22]. The MIC considered for the calculation of AUC/MIC was 1, since not all patients had positive cultures, and the majority of works consider this MIC in their models.

## 3. Statistical Analysis

A sample size calculation was performed [23], considering the removal of 20% of the drugs by the high-flux membranes during the first 2 h of the IHD session and of at least 40% at the end of the session, with an alpha error of 0.05 and a study power of 80%. A minimum samples size of 50 patients was determined.

The data obtained in the study were organized in Microsoft Excel 2007 (Microsoft Corporation, Redmond, WA, USA) sheets and posteriorly analysed with the statistical program *Sigma stat 3.5*.

Patients were compared initially according to the modality of HD (IHD vs. 6-h PHD vs. 10-h PHD). Since the best predictor of vancomycin activity is AUC/MIC and, in the literature [24], it is recommended that this ratio be maintained above 400 for the best clinical outcomes, this cut-off point was utilized to compare the patients in which it was possible to construct the 24-h AUC from clinical data.

In the descriptive statistics, data were analysed initially regarding distribution. For continuous variables, as measures of central tendency, the mean and standard deviation were used for normally distributed data, and the median and interquartile range for data with a non-normal distribution. For categorical variables, the measures of frequency and the chi-squared test were used for between-group comparisons. For comparison of continuous variables, the parametric Student’s *t*-test and ANOVA and the non-parametric Wilcoxon and Mann–Whitney tests were used. From the univariate analysis, the variables that presented *p* < 0.1 and were not co-linear were included in the multiple logistic regression analysis.

## 4. Results

From March 2015 to August 2017, 27 patients treated with IHD, 17 patients treated with PHD for 6 h and 11 patients treated for PHD for 10 h were included. 

Table 2 and Table 3 show the clinical, laboratory and dialytic characteristic of the 55 analysed patients. The mean age was 62 ± 13.12 years, with a predominance of the male sex (67.3%); the main comorbidities were systemic arterial hypertension (72.2%), diabetes mellitus (50%) and cardiovascular disease (49.1%). The most prevalent hospitalization diagnosis was sepsis (63.6%), and the lungs were the most frequent focus (67.2%). When compared with respect to their clinical characteristics, the groups were statistically significantly different regarding the prevalence of systemic arterial hypertension (*p* = 0.03), cardiovascular disease (*p* = 0.035) and sepsis as a hospitalization diagnosis (*p* = 0.02), all of which were more frequent in the IHD group when compared with the other groups.

The illness severity of the studied patients was evaluated by using the prognostic score APACHE II, which obtained a mean of 29.64 ± 7.56, and the prognosis regarding acute renal lesion by the LIANO score, which obtained a median of 0.75 (0.6–0.87). The groups differed significantly regarding APACHE II score and noradrenaline dose, which were higher in the 6-h PHD 6 and 10-h PHD groups, respectively (*p* = 0.042 and *p* < 0.001). There was no difference (*p* = 0.27) between groups with respect to mortality, the frequency of which was 74.54% in the study population.

In the 55 patients analysed, the mean dose of vancomycin was 15.51 ± 7.35 mg/kg/day, with no significant difference between dialytic treatments (*p* = 0.053). During the study period, the median number of days of vancomycin administration was 5 (2–10), with no difference between dialytic treatment groups (*p* = 0.393) and with the patients who were already in dialytic therapy for 4 days (3–7), with no difference between dialytic treatment groups (*p* = 0.152).

The distribution volume was higher and the half-life of vancomycin was longer in the PHD group when compared with the IHD group (*p* < 0.001); however, the dialytic *clearance* of vancomycin was lower in the PHD group, calculated in L/h, as described in Table 4. In the beginning of dialysis, serum concentrations of vancomycin were higher in patients treated with IHD, and this difference was maintained after 2 h (*p* < 0.001) and at the end of dialysis (*p* < 0.001). Groups were similar regarding AUC/MIC during HD but differed statistically significantly with respect to AUC/MIC at 24 h, this ratio being smaller in the 10-h PHD group (*p* = 0.047).

After 2 and 4 h of dialysis, there was no difference between groups regarding the reduction in the serum concentration of vancomycin (25.49 ± 12.49 vs. 25.32 ± 13.32 vs. 31.54 ± 11.86, *p* = 0.363 and 43.07 ± 10.06 vs. 39.28 ± 12.11 vs. 41.54 ± 12.69, *p* = 0.720, respectively). Vancomycin reduction during dialytic therapy was significantly higher in the 10-h PHD group when compared with the IHD group (43.07 ± 10.06 vs. 57.7 ± 12.13, *p* = 0.003; Figure 2). Before the dialysis session, 43 patients (78.2%) presented therapeutic vancomycin concentrations. At the end of dialysis, only 23 (41.8%) had therapeutic levels and there was a significant difference between patients treated by IHD vs. PHD (74 vs. 20%, *p* = 0.004). Of the 32 patients who had subtherapeutic concentrations, most of them were treated by PHD (68 vs. 32%, *p* = 0.004).

In order to identify the factors related to the therapeutic subconcentration of vancomycin, patients were analysed according to AUC/MIC, with a cut-off of 400, as recommended in the literature [24].

The AUC at 24 h was determined for 41 patients, 19 (46.34%) of whom presented AUC/MIC ratios less than 400. In the univariate analysis, the factors associated are highlighted in Table 5. The variables modality of dialytic therapy, distribution volume (L/kg), dialytic *clearance* (L/h) and serum concentration of vancomycin at T0 were included in the logistic regression model. The variables half-life and AUC/MIC during dialysis were excluded from the model because they were covariable with the rest. PHD was associated with AUC/MIC smaller than 400 (OR = 11.59, IC 95% 1.219–110.171, *p* = 0.033), while the higher serum concentration of vancomycin at T0 was a protective factor for AUC/MIC smaller than 400 (OR = 0.791, IC 95% 0.664–0.942, *p* = 0.009) (Table 6).

## 5. Discussion

The objective of this study was to evaluate the reduction in plasma vancomycin concentration during different dialytic modalities and to identify the factors associated with the therapeutic concentration. Although the literature about vancomycin is vast, few studies have taken this approach. Most of the existing studies about antimicrobial dosage in critically ill patients in dialytic therapy were carried out on a population treated with continuous therapy [25,26,27]. This haemodialytic method is utilized most often by nephrologists and intensivists in developed countries, on top of being a therapy that frequently utilizes convective processes, which have a high capacity to remove molecules with higher molecular weights, like vancomycin. 

On the other hand, IHD is still the main dialytic method utilized in developing countries. The majority of studies about the use of antimicrobials in IHD were realized in patients with chronic kidney disease (CKD) receiving a treatment plan of HD three times per week [28]. Although there are many studies and developed protocols, even for the CKD patients, there are still variations, controversies and inconsistencies in the literature regarding the dosage of vancomycin, involving the dosage and administration interval and the monitoring of its serum levels [28]. Even then, these recommendations are also utilized for AKI patients, despite the pharmacokinetic and pharmacodynamic alterations of the drugs used in these critically ill patients.

PHD consists of a hybrid method, with characteristics of IHD and continuous HD, and also gives adequate blood volume and metabolic control to the critically ill patient at a lower cost than continuous HD, thus being an alternative therapy to it. Studies that deal with antimicrobial dosage are still rare in this modality of HD [29,30], because they present parameters such as blood flow and dialysate and the duration of therapy as being too variable, making it difficult to design studies and protocols. Thus, generally, antibiotic dosage recommendations are utilized for IHD and CHD; however, as shown in the studies of Keough et al. [31] and Harris el al. [32], there is a great variation in the prescription of antibiotics in prolonged haemodialysis patients and the utilized dosages are frequently inadequate, possibly resulting in subtherapeutic concentrations. Lewis and Muller [33], using a mathematical model of Monte Carlo simulation applied to four different PHD prescriptions, showed that vancomycin administered at an initial dose of 15–20 mg/kg followed by 15 mg/kg after PHD would effectively reach the pharmacodynamic target in patients with *S. aureus* infections with an MIC of vancomycin ≤1 mg/L during the first 48 h of therapy. After this period, it is recommended that dosages be individualized according to the serum concentrations of vancomycin and that they follow the nomogram from the AUC/MIC in 24 h. However, those recommendations are not validated by clinical studies. The patients in our study received a dose of vancomycin of, on average, 15.51 ± 7.35 mg/kg after dialysis and, even within the doses recommended by previous studies, we found 46.34% of patients to have subtherapeutic concentrations of vancomycin (AUC/MIC <400), 84.2% of whom were receiving prolonged haemodialysis. If an MIC of 2 was utilized, practically all patients would be below the recommended value. When the modalities of HD were compared, 10-h PHD showed the lowest ASC/MIC in 24 h (504.2 (409.8–840.1) vs. 410.65 (336.34–565.39) vs. 328.53 (302.36–372.39), *p* = 0.0473), lower than what is recommended as therapeutic by the majority of the infectiology guideline*s*^26^.

When the pharmacokinetic and pharmacodynamic parameters were analysed comparing the methods of HD, PHD patients were observed to have a higher distribution volume and longer half-life. Ahern et al. [34], upon analysing the pharmacokinetics of vancomycin in prolonged haemodialysis patients, also saw a prolonged and variable half-life (43.1 ± 21.6 h) and higher distribution volume (0.84 ± 0.17 L/kg). It is possible that the findings of our work are related to a higher disease severity, characterized by higher APACHE II scores in the 6-h and 10-h PHD groups. In critical patients, due to the decrease in albumin synthesis, the passing of more protein into the extracellular space related to alterations of permeability and to uraemia itself, there is less binding of the drug to proteins, contributing to the increase in distribution volume [10] and, thus, to the higher serum concentration of vancomycin at the start of dialysis in the patients of the present study. It is important to highlight that the 6-h and 10-h PHD groups did not differ from the IHD group with respect to the administered vancomycin dose (*p* = 0.053), but there was a statistical tendency toward a higher dosage in the 10-h PHD group. PHD presented smaller dialytic *clearance* in L/h but a higher reduction in vancomycin at the end of dialysis, probably related to the longer duration of dialysis.

Despite AUC/MIC being the best predictor of the efficacy of vancomycin, there are few studies in the literature that utilize this parameter in patients undergoing renal replacement therapy, and these works focus on a treatment plan of haemodialysis for chronic renal disease [35], since *Staphylococcus aureus* infections are frequent in these patients. Thus, our study stands out by analysing the factors associated with lower AUC/MIC in 400 AKI patients in different dialytic modalities. In the multiple logistic regression, 10-h PHD was independently associated with an AUC/MIC lower than 400, while a higher serum concentration of vancomycin at the start of dialysis was a protective factor.

In contrast to previous studies that examined the removal of vancomycin during HD in critical patients, the present study included only patients in whom high permeability was used for all modalities, allowing for the evaluation of other factors such as time and dialytic *clearance*. The reduction in vancomycin at the end of therapy, calculated from the serum dosage of vancomycin at the start, at 2 h and at the end of dialytic therapy, was 43.13% for IHD, 44.23% for 6-h PHD and 57.70% for 10-h PHD, *p* = 0.042. Our results were higher than those found by Golestaneh et al. [36], which presented a reduction of 36% in 10 patients receiving PHD with a duration of at least 8 h. When they analysed the removal of vancomycin from dialysate in patients with sepsis and AKI treated with PHD using high-flux membranes, Kielstein et al. [37] observed 26% removal of vancomycin from the dialysate. In a study by our group, the removal of vancomycin through high-volume peritoneal dialysis was 21.7% [38].

Our study presents some limitations. Its design was cross-sectional, since patients were analysed in a single moment. It was realized in a single centre, including only 55 patients. A backfire effect was not considered, given by the redistribution of vancomycin from the tissues to the bloodstream, which may lead to an increase of 30–40% in serum levels of vancomycin 4–6 h after the end of intermittent haemodialysis [39]. This effect is lower in PHD, due to the higher duration of dialysis, resulting in a higher distribution of the drugs through the tissues along the therapy. Vancomycin was not dosed in the dialysate for the calculation of its removal by dialytic therapy, because, on top of the increased costs that this would have incurred, our study included only high-flux membranes and evaluated different durations of PHD with the goal of identifying the factors associated with higher reduction and therapeutic concentrations. Moreover, the obtained results serve as orientation for new studies and proposals of antimicrobial dosage in critically ill patients, those with AKI and undergoing renal replacement therapy, which are difficult to evaluate from the point of view of drug pharmacokinetics and pharmacodynamics, since they are subjected to various alterations, many of which are still not fully understood in the literature. 

## 6. Conclusions

Our study in septic AKI patients treated by HD with high-flux membranes showed a high prevalence of patients that do not reach therapeutic concentrations, especially in the population treated by PHD. Such results may be related to a higher risk of bacterial resistance and mortality, besides pointing out the necessity of additional doses of vancomycin during prolonged haemodialysis and not only at the end of therapy as realized in clinical practice. However, future studies are necessary to evaluate vancomycin dosage protocols in critical patients in IHD or PHD that have the objective of guaranteeing therapeutic concentrations. These will be of great relevance for developing countries, where such dialytic methods are more often utilized, and serum analysis of vancomycin for the adjustment of the drug is not always available.

## Figures and Tables

**Figure 1 ijerph-17-06861-f001:**
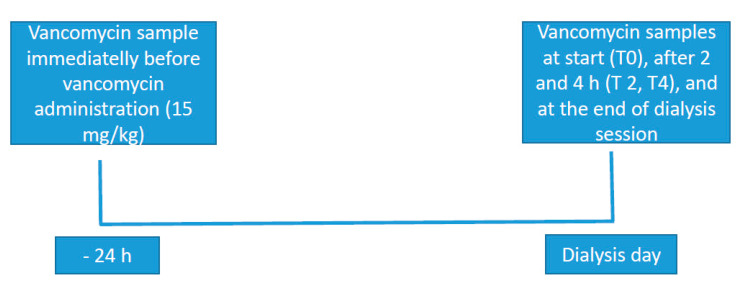
Dialysis procedure, administration of vancomycin and sampling.

**Figure 2 ijerph-17-06861-f002:**
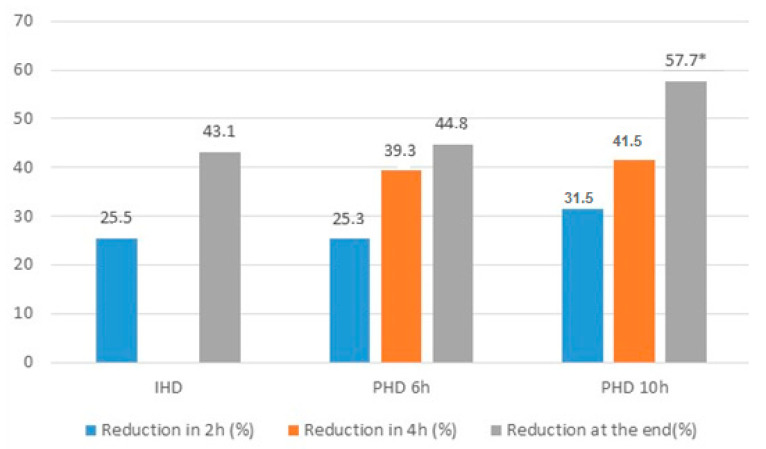
Reduction of vancomycin in 2 h, 4 h and at the end of dialysis in the groups HDI, HDP 6 h and HDP 10 h. * *p* < 0.05 PHD 10 h vs. IHD, *p* < 0.05.

**Table 1 ijerph-17-06861-t001:** Formulas utilized for the calculation of Vancomycin pharmacokinetic [18,19,20].

Pharmacokinetic	Formula
Distribution Volume (L)	VD = Vancomycin dose (mg)/concentration of vancomycin (mg/L)
Dialytic *Clearence* of vancomycin (L/h)	Cl vanco = (Initial—final concentration)/Initial concentration × QD/t
Half-life time	T½ life = 0.693 × VD/Cl
% reduction in vancomycin	% red = (C pre HD—C final HD)/C pre HD) × 100

VD: Distribution volume. Cl vanco: Dialytic *Clearence* of vancomycin. QD: Dialysate fluxo (L/h). t: time h). T½ life: half-life time. C pre HD: concentration of vancomycin pre hemodialysis. C final HD: Concentration of vancomycin at the end of hemodialysis.

**Table 2 ijerph-17-06861-t002:** Clinical characteristics of the general population and of the groups according to the different dialytic treatments.

Variables	General (*n* = 55)	IHD (*n* = 27)	PHD 6 h (*n* = 17)	PHD 10 h (*n* = 11)	*p*
Age (Years)	62.61 ± 13.12	63.30 ± 13.86	61.94 ± 14.15	62 ± 10.36	0.934
Male sex (%)	37 (67.3)	17 (62.86)	13 (76.47)	6 (54.54)	0.457
Weight (kg)	75.07 ± 24.20	71 (64–84.75)	69 (57.32–77.75)	62 (58.50–85.75)	0.594
Comorbidities					
Diabetes (%)	28 (50.0)	13 (48.15)	10 (58.82)	5 (45.45)	0.726
Hypertension (%)	40 (72.72)	24 (88.89) ^a^	10 (58.82) ^b^	6 (54.54) ^b^	0.030
Cardiovascular disease (%)	27 (49.09)	18 (66.67) ^a^	5 (29.41) ^b^	4 (36.36) ^ab^	0.035
Obesity (%)	9 (16.36)	3 (11.11)	2 (11.76)	4 (36.36)	0.134
Neoplasia (%)	6 (10.90)	2 (7.41)	2 (11.76)	2 (18.18)	0.621
Smoking (%)	25 (45.45)	11 (40.74)	8 (47.06)	6 (54.54)	0.731
Chronic Kidneyl Disease (%)	4 (7.27)	2 (7.40)	1 (5.88)	1 (9.09)	0.950
Other (%)	30 (54.54)	13 (48.15)	10 (58.82)	7 (63.64)	0.626
Hospitalization Diagnostic					
Sepsis (%)	35 (63.64)	22 (81.48) ^a^	7 (41.18) ^b^	6 (54.54) ^b^	0.02
Acute abdomen (%)	7 (12.73)	3 (11.11)	3 (17.65)	1 (9.09)	0.754
Acute Coronarary Syndrome (%)	6 (10.90)	5 (18.6)	1 (5.88)	0	0.183
Trauma (%)	3 (5.45)	1 (3.70)	1 (5.88)	1 (9.09)	0.799
Elective Surgery (%)	7 (12.73)	2 (7.41)	3 (17.65)	2 (18.18)	0.508
Other (%)	3 (5.45)	1 (3.70)	0	2 (18.18)	0.100
Arterial Thrombosis (%)	7 (12.73)	5 (18.52)	2 (11.76)	0	0.172
Infectious Focus					
Lung (%)	37 (67.27)	21 (77.78)	9 (52.94)	7 (63.64)	0.223
Urinary (%)	3 (5.45)	2 (7.41)	1 (5.88)	0	0.657
Abdominal (%)	6 (10.90)	1 (3.70)	3 (17.65)	2 (18.18)	0.242
Cutaneous (%)	5 (9.09)	1 (3.70)	1 (5.88)	3 (27.27)	0.062
Other (%)	6 (10.90)	3 (11.11)	3 (17.65)	0	0.343
APACHE II	29.64 ± 7.56	27.16 ± 7.930 ^a^	33.06 ± 7.07 ^b^	30.00 ± 5.59 ^ab^	0.042
LIANO	0.75 (0.6–0.87)	0.72 (0.54–0.78)	0.78 (0.71–0.84)	0.89 (0.44–0.89)	0.163
Death (%)	41 (74.54)	19 (70.37)	15 (88.23)	7 (63.63)	0.27

^a^ significantly different from ^b^. ^ab^: similar to ^a^ and ^b^. IHD: conventional intermittent dialysis. PHD 6 h: Hemodialysis prolonged for 6 h. PHD 10 h: Hemodialysis prolonged for 10 h.

**Table 3 ijerph-17-06861-t003:** Laboratory and dialytic characteristics of the general population and of the groups according to different dialytic treatments on the day of the study.

Variables	General (*n* = 55)	IHD (*n* = 27)	PHD 6 h (*n* = 17)	PHD 10 h (*n* = 11)	*p*
Noradrenaline (mcg/kg/min)	0.25 (0.05–0.5)	0.05 (0.0–0.14) ^a^	0.5 (0.33–0.8) ^b^	0.45 (0.38–0.57) ^b^	<0.001
Urinary Volume (mL/24 h)	200 (50–650)	300 (133–950)	150 (27.5–800)	100 (0.0–260)	0.072
Hematocrit (%)	27.85 ± 3.89	26.5 (23.8–28.9)	27.9 (26.75–30.35)	27.8 (26.8–29.9)	0.062
PCR (mg/L)	27.1 (7.3–33.7)	25.8 (5.50–31.82)	23.2 (8.12–35.3)	27.8 (23.8–35.1)	0.518
Albumin (g/dL)	2.18 ± 0.49	2.21 ± 0.43	2.17 ± 0.51	2.11 ± 0.67	0.049
Ultrafiltration (mL)	1901.82 ± 1148.19	2018.52 ± 1138.84	1564.71 ± 1225.22	2136.36 ± 1026.91	0.339
Kt/V	0.89 ± 0.33	0.86 ± 0.28	0.82 ± 0.35	1.08 ± 0.36	0.093
Intradialitic hypotension	9 (16.36)	4 (14.81)	4 (23.52)	1 (9.09)	0.574

^a^ significantly different from ^b^. IHD: conventional intermittent dialysis. PHD 6 h: Hemodialysis prolonged for 6 h. PHD 10 h: Hemodialysis prolonged for 10 h. PCR: reactive protein C.

**Table 4 ijerph-17-06861-t004:** Characteristics associated with the prescription of vancomycin and to the pharmacodynamic and pharmacokinetic of the drug in the general population and on the groups according to different dialytic treatments on the day of the study.

Variables	General (*n* = 55)	IHD (*n* = 27)	PHD 6 h (*n* = 17)	PHD 10 h (*n* = 11)	*p*
Vancomycin days	5 (2–10)	7 (3–12)	5 (2–6.5)	4 (2–12)	0.39
Vancomycin dose (mg/kg/dia)	15.5 ± 7.3	13.9 (8.6–15.4)	14.7 (13.5–18.0)	18.7 (12.0–24.6)	0.05
Vancomycin concentration before analysis (mg/L)	19.6 (16–27.8)	20.9 (17.8–31.5)	20.2 (15.4–27.7)	17.9 (14.1–23.0)	0.61
Distribution volume (L/Kg)	0.6 (0.4–0.85)	0.40 (0.29–0.62) ^a^	0.8 (0.61–1.0) ^b^	0.7 (0.59–1.29) ^b^	<0.001
Half-life time (h)	17.4 (6.7–32.6)	6.70 (4.2–11.9) ^a^	27.6 (19.5–74.4) ^b^	38.4 (30.62–74.78) ^b^	<0.001
Vancomycin dialytic *Clearence* (L/h)	1.9 (1.13–3.23)	3.2 (2.54–3.68) ^a^	1.3 (0.93–1.72) ^b^	1.0 (0.75–1.15) ^b^	<0.001
Vancomycin concentration (mg/L) at T0	26.0 (15.9–35.9)	34.2 (22–44) ^a^	13.4 (10.7–18.4) ^b^	23.1 (13.6–31.0.6) ^ab^	<0.001
Vancomycin concentration at T 2 h (mg/L)	17.7 (11.39–28.2)	22.8 (17.45–33.04) ^a^	11.1 (8.36–15.26) ^b^	16.6 (9.45–19.03) ^b^	<0.001
Vancomycin concentration at T final (mg/L)	14.2 (8.7–20.9)	20.26 ± 9.91 ^a^	11.32 ± 5.71 ^b^	10.6 ± 5.96 ^b^	<0.001
% of reduction in 2 h	26.6 ± 12.6	25.5 ± 12.5	25.3 ± 13.3	31.5 ± 11.8	0.36
% Total reduction	45.7 ± 12.79	43.1 (33.2–49.8) ^a^	44.7 (32.9–57.9) ^ab^	57.7 (40.5–64.3) ^b^	0.03
AUC/MIC 24 h	411.37 (326.64–580.17)	504.2 (409.8–840.1) ^a^	410.65 (336.34–565.39) ^ab^	328.53 (302.36–372.39) ^b^	0.04
AUC/MIC during analysis	138.0 (99.6–215.2)	169.8 (112.5–235.3)	103.0 (94.8–178.1)	124.7 (92.8–185.1)	0.13

^a^ significantly different from ^b^. ^ab^: similar to ^a^ and ^b^. IHD: conventional intermittent dialysis. PHD 6 h: Hemodialysis prolonged for 6 h. PHD 10 h: Hemodialysis prolonged for 10 h. T final: final time of dialysis. AUC/MIC: area under the curve/minimum inhibitory concentration.

**Table 5 ijerph-17-06861-t005:** General characteristics of the population according to the area under the curve/minimum inhibitory concentration of Vancomycin.

Variables	General Population (*n* = 41)	AUC/MIC 24 h <400 (*n* = 19)	AUC/MIC 24 h >400 (*n* = 22)	*p*
Age (years)	65.00 (56.50–72.50)	65.00 (57.00–74.00)	65.00 (53.75–72.25)	0.724
Male sex (%)	15 (63.41)	14 (73.68)	12 (54.54)	0.345
Weight (kg)	70.00 (60.50–82.50)	71.00 (62.00–86.00)	70.00 (54.25–74.25)	0.36
APACHE II	32.00 (26.00–36.00)	32.47 ± 6.04	28.91 ± 7.79	0.114
LIANO	0.77 (0.61–0.89)	0.83 ± 0.72	0.75 ± 0.51	0.123
Noradrenaline (mcg/kg/min)	0.33 (0.06–0.53)	0.38 (0.20–0.57)	0.15 (0.00–0.52)	0.11
Urinary volume (ml/24 h)	200.00 (40.00–425.00)	100.00 (25.00–450.00)	200.00 (45.00–500.00)	0.591
Hematocrite (%)	28.10 ± 3.87	27.39 ± 3.75	28.71 ± 3.95	0.281
PCR (mg/L)	24.80 (6.60–33.65)	25.78 ± 14.67	19.12 ± 14.30	0.15
Albumin (g/dL)	2.18 ± 0.49	2.14 ± 0.51	2.23 ± 0.48	0.587
Conventional hemodialysis (%)	16 (39.02)	3 (15.79)	13 (59.10)	0.012
Prolonged hemodialysis 6 and 10 h (%)	25 (60.98)	16 (84.21)	9 (40.90)	0.012
Kt/V	0.89 ± 0.33	0.90 ± 0.35	0.88 ± 0.31	0.814
Dose of Vancomycin (mg/kg/day)	14.71 (12.69–19.37)	14.71 (10.71–20.00)	14.60 (13.29–17.86)	0.734
Volume of distribution (L/Kg)	0.67 (0.45–0.87)	0.73 (0.61–1.34)	0.56 (0.34–0.83)	0.018
Half-life time	19.67 (9.36–38.14)	28.38 (17.51–79.69)	11.22 (5.99–26.28)	0.004
Vancomycin dialytic clearance(L/h)	1.69 (1.03–3.05)	1.19 (0.95–1.70)	2.39 (1.15–3.55)	0.017
Vancomycin concentration (mg/L) at T0	23.59 (14.92–33.87)	14.96 (13.19–24.20)	32.83 (21.47–37.95)	0.001
% of reduction in 2 h	26.52 ± 13.29	27.18 ± 12.23	25.94 ± 14.41	0.771
% Total reduction	46.98 ± 13.05	50.56 ± 14.62	43.90 ± 10.95	0.104
AUC/MIC during dialysis	120.44 (94.85–211.45)	103.07 (87.35–169.25)	172.43 (108.01–252.98)	0.016
Death (%)	30 (73.17)	16 (84.21)	14 (63.64)	0.259

AUC/MIC: area under the curve/minimum inhibitory concentration.

**Table 6 ijerph-17-06861-t006:** Logistic regression of the variables associated to AUC/MIC < 400.

Variables	Odds Ratio	Confidence Interval	*p*
Prolonged hemodialysis	11.59	1.219–110.171	0.033
Distribution volume (L/Kg)	0.197	0.0123–3.153	0.251
Dialytic clearance of vancomycin (L/h)	1.614	0.391–6.672	0.508
Serum concentration of vancomycin (mg/L) at T0	0.791	0.664–0.942	0.009

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
