# Peer review of "Vancomycin for Dialytic Therapy in Critically Ill Patients: Analysis of Its Reduction and the Factors Associated with Subtherapeutic Concentrations"

_ijerph, 2020, doi:10.3390/ijerph17186861_

Round 1
Reviewer 1 Report
This is a well written article. There are very few grammatical errors.
In the title, "Use" of vancomycin will need to be changed as the authors are investigating vancomycin levels in patients on dialysis and not investigating levels for dialysis.
Please change Acute "Coronarian" syndrome to coronary in Table 2.
Please change Laboratorial to Laboratory in Table 3.
Please remove "of" from "43 of patients" in line 191.
Author Response
Dear reviewer,
Thank you very much for your comments and suggestion. They were done.
We have changed the title, Acute "Coronarian" syndrome to coronary in Table 2, Laboratorial to Laboratory in Table 3 and we have removed "of" from "43 of patients" in line 19.
Thanks a lot.
Sincerely,
Daniela Ponce

Reviewer 2 Report
Freitas et al. present an excellent paper exploring the less reasearched aspects of vancomycin dosing in crtically ill patient requring dialysis - in particular those who are septic. From their prospective cross-sectional cohort, they note that patients who undergo prolonged haemodialysis have sub-optimal concentrations of vancomycin dosing and this may impact upon bacterial resistance and prognosis - although direct evidence for the latter two has not been investigated in this study.
The introduction and methods are well described. I would say that the results section could be improved - too much information is provided in the tables.
This study provides future directions for further study and will be useful reading for those units who use prolonged durations of renal replacement therapy e.g. ICU settings.
Author Response
Dear reviewer,
Thank you very much for considering our study interesting and well written. WE have tried to become the tables clearer.
Sincerely,
DP

Reviewer 3 Report
This is a nice paper. However, I have some comments.
The findings from this paper are excellent and worthy to review.
This manuscript contained some questions described below.
I think this paper is interesting, this review contributes to future's clinical medicine largely. I have some questions from a point of view of clinical medicine.
In conclusion, does the removal rate of vancomycin by hemodialysis have the most effect on dialysis time? Does it mean that efficient dialysis in a short time can best lead to the effect of vancomycin?
However, patients in the long-term dialysis group are presumed to have unstable circulatory and respiratory kinetics due to worsening of general condition, so it is problem that the dialysis time alone is a problem. There is no significant difference in APACHE II score, so it cannot be said that there is no problem.
We think that it is difficult to compare the two groups because of difference of the backgrounds of patients. Is it a question that the long-term dialysis group has few cardiovascular complications?
Hypoalbuminemia is conspicuous overall, but is it due to undernutrition due to sepsis, or are there many patients who are originally malnourished?
Many patients who are diagnosed with sepsis are in the short-term dialysis group, but patients who are diagnosed with sepsis are likely to have poorer general condition, so more frequent long-term dialysis will be performed. How do you explain this difference?
Patients who need to receive vancomycin will be those diagnosed with sepsis. By method, the target patients are described as patients aged 18 years or older with acute tubular necrosis associated with sepsis. What are the criteria for selecting the long-term dialysis group?
If long-term dialysis is selected due to deterioration of general condition, continuous hemofiltration dialysis should be the standard. In that case, a hemofilter will be used. How will the removal rate of vancomycin change in that case? How does the vancomycin removal rate change for ST69 membranes as well?
To what extent does the patient population include chronic maintenance dialysis patients? Finally, please tell us the optimal vancomycin administration method that can be considered from this study.
Author Response
Dear reviewer 3 (R), thank you very much for your questions and suggestion.
We, authors (A), tried to be clearer.
Sincerely,
R: This is a nice paper. However, I have some comments.
The findings from this paper are excellent and worthy to review.
This manuscript contained some questions described below.
I think this paper is interesting, this review contributes to future's clinical medicine largely. I have some questions from a point of view of clinical medicine.
In conclusion, does the removal rate of vancomycin by hemodialysis have the most effect on dialysis time? Does it mean that efficient dialysis in a short time can best lead to the effect of vancomycin?
A: This study showed greater removal of vancomycin in patients at PHD 6 and 10h when compared to HDI patients and, consequently, these groups (PHD 6 and 10h) had a higher frequency of patients with AUC / MIC below 400. So, really the factor therapy time seems to be very important, since there was no difference in the efficiency of dialysis between these groups, a factor assessed by the measurement of Kt / V, according to tables 3 and 5.
R: However, patients in the long-term dialysis group are presumed to have unstable circulatory and respiratory kinetics due to worsening of general condition, so it is problem that the dialysis time alone is a problem. There is no significant difference in APACHE II score, so it cannot be said that there is no problem.
We think that it is difficult to compare the two groups because of difference of the backgrounds of patients. Is it a question that the long-term dialysis group has few cardiovascular complications?
A: The prescription of conventional or prolonged hemodialysis was defined by the patient's hemodynamic conditions on the day the dialysis was analyzed. Patients without the use of vasoactive drugs or those with noradrenaline up to 0.5 mcg / kg / min were dialysed by conventional therapy, while those with noradrenaline greater than 0.5 mcg / kg / min were treated with prolonged hemodialysis. The duration of 6 or 10 hours was determined by the nephrologist according to clinical evaluation in relation to hemodynamic instability, hydric balance and nitrogen scoria. Thus, the most severe patients tend to be dialysed for a longer time. The APACHE II score, used to assess patient severity, in general, was higher in patients undergoing prolonged hemodialysis.
When comparing the groups (IHD vs. PHD 6h vs. PHD 10h), a higher frequency of cardiovascular disease was observed in the IHD group (Table 2). This finding was not considered relevant for our analysis, since our study did not it was prospective and did not assess mortality. As the study was cross-sectional with the objective of evaluating the kinetics of vancomycin in patients receiving dialysis, this difference does not affect our results.
In our study, the lower prevalence of cardiovascular disease as a comorbidity in the PHD group was not expected, since it was the patients who evolved with greater severity and greater need for vasoactive drugs. Cardiovascular complications during hospitalization were not evaluated.
R: Hypoalbuminemia is conspicuous overall, but is it due to undernutrition due to sepsis, or are there many patients who are originally malnourished?
A: The serum albumin (table 3) was measured on the day the patient was analyzed in the study. Thus, as we do not have the albumin of these patients before hospitalization, there is no way to measure the frequency of undernutrition previous to sepsis.
R: Many patients who are diagnosed with sepsis are in the short-term dialysis group, but patients who are diagnosed with sepsis are likely to have poorer general condition, so more frequent long-term dialysis will be performed. How do you explain this difference?
A:In Table 2, the diagnosis of sepsis considered was at the time of admission, thus, there are patients who were hospitalized for another reason and developed sepsis at some point during hospitalization, when they were included in our study. Thus, it may be that patients who evolved with nosocomial infection presented with more severe sepsis, requiring prolonged therapy. The hypothesis is that infection by hospital germs in an individual who is already compromised by hospitalization, may develop with more severe sepsis.
R: Patients who need to receive vancomycin will be those diagnosed with sepsis. By method, the target patients are described as patients aged 18 years or older with acute tubular necrosis associated with sepsis. What are the criteria for selecting the long-term dialysis group?
A: We have answered above.
R: If long-term dialysis is selected due to deterioration of general condition, continuous hemofiltration dialysis should be the standard. In that case, a hemofilter will be used. How will the removal rate of vancomycin change in that case? How does the vancomycin removal rate change for ST69 membranes as well?
A: In the literature, studies of vancomycin pharmacokinetics in patients on continuous therapy are more frequent (references 27-29), because in addition to the time factor, the hemofilters used have a greater capacity to remove large molecules. Thus, as the removal of vancomycin in this dialysis method is very important, it is already established in the literature to correct the dose of Vancomycin for a creatinine clearance of 50ml / min and the doses and administration intervals adjusted according to the serum dosage of vancomycin, as described in Sanford Guide to antimicrobial therapy.
R: To what extent does the patient population include chronic maintenance dialysis patients? Finally, please tell us the optimal vancomycin administration method that can be considered from this study.
A: Patients with chronic kidney disease in hemodialysis prior to hospitalization were not included in the study.
A:This study showed that PHD patients, especially the 10h group, had subtherapeutic vancomycin concentrations during and at the end of therapy. This suggests that patients on this modality of dialysis should receive an additional dose of Vancomycin during therapy, however further studies are needed to determine the dose and method of administration (continuous or bolus infusion).
